# Progress in Salicylic Acid-Dependent Signaling for Growth–Defense Trade-Off

**DOI:** 10.3390/cells11192985

**Published:** 2022-09-25

**Authors:** Ching Chan

**Affiliations:** Department of Life Science, National Taiwan Normal University, Taipei 11677, Taiwan; chan.ching@ntnu.edu.tw

**Keywords:** plant immunity, salicylic acid, phytohormones, growth–defense trade-off

## Abstract

One grand challenge for studying plant biotic and abiotic stress responses is to optimize plant growth and plasticity under variable environmental constraints, which in the long run benefits agricultural production. However, efforts in promoting plant immunity are often accompanied by compromised morphological “syndromes” such as growth retardation, sterility, and reduced yield. Such a trade-off is dictated by complex signaling driven by secondary messengers and phytohormones. Salicylic acid (SA) is a well-known phytohormone essential for basal immunity and systemic acquired resistance. Interestingly, recent updates suggest that external environmental cues, nutrient status, developmental stages, primary metabolism, and breeding strategies attribute an additional layer of control over SA-dependent signaling, and, hence, plant performance against pathogens. In this review, these external and internal factors are summarized, focusing on their specific roles on SA biosynthesis and downstream signaling leading to immunity. A few considerations and future opportunities are highlighted to improve plant fitness with minimal growth compensation.

## 1. Introduction

In nature, plants are continuously challenged by a mix of environmental conditions, including temperature fluctuation, elevated atmospheric CO_2_, drought, flooding, saline soil, a variety of pathogens and insect herbivores. To cope with the simultaneous exposure to combined external variations, plants have evolved inducible mechanisms to balance assimilates and signaling molecules between growth and defense. Such a trade-off is believed to be primarily due to resource restrictions [1]. The regulation network relies on several common signaling pathways, including phytohormone balance, the production of reactive oxygen species (ROS), calcium signatures, the activation of kinase cascades, and sugar signals [1,2] to ensure robust defense response with minimum fitness cost. How plants integrate these inducible mechanisms to cope with combined biological and physical stimuli and to balance growth and defense remains largely unexplored.

Recent reports suggest that plant responses to combined stresses might not simply be additive of individual stresses [3,4]. Meta-analysis indicates that gene expression under multifactorial stress is not predictable from single stress exposures [5,6], arguing for investigating plant responses to different combinations of biotic and abiotic stresses. However, different laboratory conditions and variation in plant species, pathogens, treatment time, and strength lead to discrepancies between published works. Nevertheless, some independent reports do converge on common phytohormone signaling or central metabolic pathways, highlighting the significance of these important hubs, which form the foundation for further exploration. In general, phytohormones such as salicylic acid (SA), jasmonic acid (JA), and ethylene (ET) are considered major regulators of plant defense responses. Conventionally, SA signaling is induced by and involved in defense against biotrophic and hemi-biotrophic pathogens and viral infection, as well as for establishing systemic acquired resistance [7,8]. On the other hand, JA/ET signaling is activated by necrotrophic pathogens and insect herbivores [9]. For combined stresses, the signature and corresponding transcriptional changes might be difficult to evaluate and will be a future challenge. In this review, we summarize current updates on the impact of several environmental factors on plant immunity, focusing on their roles in modulating the SA-dependent signaling pathway. This information might provide a framework to further explore the signaling network underlying the growth–defense trade-off (Figure 1) and in the long run to translate laboratory findings to field application.

## 2. SA Biosynthesis, Perception, and Signaling in Defense and Growth

SA is an essential phytohormone for basal immunity and systemic acquired resistance (SAR) against biotrophic and hemi-biotrophic pathogens. The biosynthesis of SA has been thoroughly described in excellent reviews [7,8]. Briefly, SA is synthesized by two independent pathways using chorismate or phenylpropanoid as the precursors. The isochorismate synthase (ICS) pathway is the major process that mediates SA induction upon pathogen infection. In the Arabidopsis (*Arabidopsis thaliana*) genome, two isochrismate synthase genes (*ICS1* and *ICS2*) encode the enzymes that convert chorismate to isochrismate (IC) for initiating SA production in the chloroplast. Mutation of *ICS1* (*ics1*, or alternatively *SA-deficient 2*, *sid2* mutant) hampers up to 90% SA induction upon pathogen infection, indicating that *ICS1* plays a major role over *ICS2* in defense. IC in the chloroplast is transported to the cytosol by the MATE transporter enhanced disease susceptibility 5 (EDS5) [10,11,12]. Finally, avrPphB susceptible 3 (PBS3) catalyzes the conjugation of IC to glutamate (Glu), producing IC9-Glu, which spontaneously breaks down in the cytosol to produce SA [10,13]. *ICS1*, *EDS5*, and *PBS3* are strongly induced upon pathogen infection. Glucose conjugation of SA produces SA glucoside (SAG), which is actively transported from the cytosol into the vacuole as an inactive storage form. Although the ICS pathway is seemingly important for SA production, the alternative pathway—the phenylalanine ammonia lyase (PAL) pathway—is also indispensable. Quadruple mutation of Arabidopsis *PAL* genes showed 75% SA reduction at rest and 50% SA reduction upon pathogen infection [14]. The large reduction of pathogen-induced SA level when either the ICS or PAL pathway is disrupted is puzzling. How ICS and PAL pathways interplay to maintain the SA pool remains to be established.

Identification of cellular components for SA perception has mainly been carried out by forward genetic screening. To date, the most well-documented SA binding proteins are the nonexpressor of PR genes (NPRs), including NPR1, NPR3 and NPR4, which serve as bona fide SA receptors [15]. Recombinant NPR1 binds to SA with a dissociation constant (Kd) range from 140 to 220 nM in vitro [15,16]. NPR1 is the master regulator for SA-induced PR gene expression and, hence, resistance against pathogens. Mutation of *NPR1* disables *PR* gene expression and response to various SAR-inducing treatments and promotes susceptibility to infections [17]. NPR1 protein consists of an N-terminal BTB/POZ domain, a central Ankyrin repeat (ANK) region, and a C-terminal transactivation domain [17,18]. Interestingly, NPR1 does not bear a DNA-binding domain. Therefore, NPR1 is considered a transcriptional co-activator that depends on physical interaction with a partner transcription factor, such as WRKYs and TGAs, for activating downstream SA-responsive genes.

SA also plays an important role in plant growth and development [19]. In general, SA over-accumulation negatively affects plant growth. Exogenous SA application inhibits vegetative growth in soybean (*Glycine max*) [20], wheat (*Triticum aestivum*) [21], maize (*Zea mays*) [22], and chamomile (*Matricaria chamomilla*) [23]. Many autoimmune mutants have been reported, and many of them exhibit SA over-accumulation and dwarfism [24]. Notably, the growth inhibition in these autoimmune mutants can be rescued by blocking SA biosynthesis or SA signaling, reiterating the negative effect of SA on plant growth. Thus, mutation of SA biosynthetic enzyme (*sid2* mutant) or disrupting SA accumulation (by introducing *nahG* transgene) leads to increased biomass and seed yield [25].

## 3. Environmental Conditions Modulate SA Biosynthesis and SA Signaling

Elevated temperature and atmospheric CO_2_ are two major environmental conditions due to global climate change. The impact of these environmental changes on defense signaling appears to be indirect and pleotropic because they lead to changes in cellular processes with overlapping functions and basic physiological programs. However, increasing evidence suggests that the growth–defense trade-off process is adaptive and specific even though the two processes share similar signaling molecules and use overlapping signaling cascades.

### 3.1. Temperature

Temperature sensitivity in plant disease resistance was first recognized in the 1900s [26,27]. The impact of elevated (and low) temperature on plant disease resistance was then tested in many plant–pathogen systems. In general, there seems to be a negative correlation between elevated temperature and disease resistance in plants.

Early studies showed that low temperature promoted the accumulation of SA and SAG in Arabidopsis, accompanied by upregulation of *ICS1*, *CBP60g*, and *SARD1* transcripts [28]. The cold-stress-induced bacterial resistance phenotype was eventually confirmed by challenging Arabidopsis with the bacterial pathogen *Pseudomonas syringae* pv. *tomato* strain DC3000 (*Pst* DC3000) [29,30]. Bacteria titer was significantly reduced using 4 °C [29] or 16 °C [30] treatment conditions. Notably, an intact SA signaling pathway is required for cold-stress-induced resistance. *Sid2* mutation [30] and *nahG* transgene [29] abolished cold-stress-induced protection. Interestingly, mutation of *NPR1* only partially abolished cold-stress-induced protection, indicating that an NPR1-independent pathway exists.

On the other hand, elevated temperature promotes virus susceptibility in Arabidopsis [3], tomato (*Solanum lycopersicum*) [31,32], and potato (*Solanum tuberosum*) [33]. SA subverts heat-stress-induced viral sensitivity in potato cultivars. In *S. tuberosum* L. cv. Gala, *StPR1-b* gene expression was rapidly induced upon potato virus Y (PVY) infection at both 22 °C and 28 °C. However, in *S. tuberosum* L. cv. Chicago, *StPR1-b* gene expression was only induced at 22 °C but not 28 °C. Consistently, PVY RNA accumulation, as determined by qRT-PCR, revealed that Gala was more resistant, while the viral load in Chicago was up to 20-fold higher [33]. In addition, *N* gene conferred gene-for-gene resistance to the viral pathogen tobacco mosaic virus (TMV). Two independent studies confirmed that the hypersensitive response triggered by TMV was heat-dependent in tomato (*S. lycopersicum*) [31] and tobacco (*Nicotiana benthamiana*) [34]. Under elevated temperature, N protein showed reduced nuclear accumulation, presumably due to conformational change [35]. Similarly, heat-dependent suppression of disease resistance has also been reported in plant–bacteria interactions. At 28 °C, Arabidopsis plants showed more severe disease symptoms and increased bacterial growth when challenged with virulent *Pst* DC3000 [34]. However, *sid2* mutation and *nahG* transgenic plants retained temperature sensitivity, indicating that the involvement of SA signaling was minimal in this case study [34].

### 3.2. Atmospheric CO_2_

The other important variant as a result of global climate change is elevated atmospheric CO_2_ (eCO_2_), which can modulate the balance between hormone levels in many plant species [36]. The impact of eCO_2_ on biological process besides primary metabolism includes biotic stress responses, as has been revealed by transcriptomic studies in Arabidopsis [37] and wheat (*T. aestivum*) [38]. Quantification of defense hormones further confirmed the observation. Elevated CO_2_ induced SA accumulation in Arabidopsis [39,40], beans (*Phaseolus vulgaris*) [40], wheat (*T. aestivum*) [40], and tomato (*S. lycopersicum*) [41]. The associated induction of SA marker gene *PR1* was stronger under eCO_2_ in different plant species [39,40,41]. Consistently, the plants showed enhanced resistance against the biotrophic oomycete *Hyaloperonospora arabidopsidis* [39] and *Pst* DC3000 [40,41]. For instance, an intact NPR1-dependent pathway was required for bacteria resistance in tomato [41], highlighting the specificity of an SA-dependent pathway in immune signaling mediated by eCO_2_. However, in Arabidopsis *sid2* and *npr1* mutants, eCO_2_-induced resistance against oomycete was only partially affected [39], underlying the involvement of an alternative pathway other than the priming of SA-dependent defense. On the other hand, the opposite effect of eCO_2_ has also been reported in other studies using different plant species and pathogen. In maize (*Z. mays*), eCO_2_ did not significantly induce SA nor JA levels under unchallenged conditions [42]. However, after infection with the fungal pathogen *Fusarium verticillioides*, JA induction was abolished while SA level was repressed. As a result, eCO_2_ negatively affected fungal resistance in maize [42]. In wheat (cv. *Remus*), eCO_2_ was shown to enhance virulence of the fungal pathogen *Zymoseptoria tritici* and hence promoted plant susceptibility [43].

Intriguingly, the antagonism between SA and JA was not consistent between Arabidopsis and tomato. Under eCO_2_, both SA and JA levels were significantly induced in Arabidopsis [39]. The JA-inducible gene *VSP2* was upregulated by ~400-fold, and the plants were more resistant to the necrotrophic fungus *Plectosphaerella cucumerina* [39]. A subsequent report by Zhou et al. (2019) [44] also detected upregulation of the JA marker gene *PDF1.2* under eCO_2_, and enhanced resistance against the necrotrophic fungus *Botrytis cinerea*. However, in the same study, eCO_2_ suppressed SA-signaling and reduced plant resistance against *Pst* DC3000, which was opposite to the observation by Williams et al. (2018) [39]. On the other hand, JA level was not significantly affected by eCO2 in tomato, and the plants were more susceptible to *B. cinerea* infection [41]. The discrepancy is likely due to a different experimental setup, as eCO_2_ significantly affects plant development and canopy density [45,46], along with differences in other variables such as plant growth conditions and the strength and duration of pathogen treatment.

From an evolutionary point of view, sub-ambient CO_2_ (saCO_2_) provides another direction to evaluate the crosstalk between atmospheric CO_2_ and plant immunity, such as what might have occurred in pre-industrial or glacial periods [47,48]. Transcriptomic study revealed that saCO_2_ promotes photorespiratory processes [49] that correlate with pathogen defense [50]. Photorespiration and peroxisomal metabolism are the major sources of intracellular reactive oxygen species (ROS), which are also among the early signals of pathogen infection. Loss of CATALASE, a ROS scavenging enzyme, promoted constitutive defense [51], while loss of respiratory burst oxidase homologues (RBOHs) caused ROS accumulation, SA induction, and Arabidopsis resistance against *Pst* DC3000 [52]. How saCO_2_/eCO_2_-mediated alteration in defense interplays with key players in ROS signaling remains to be addressed.

It has been proposed that atmospheric CO_2_ concentration, which determines plant photosynthetic rate, might also impact carbon relocation and, hence, general plant development. The CO_2_ “fertilization effect” has been long known for promoting plant growth by increasing photosynthesis, C:N ratio, and water-use efficiency [53]. This can impose an additional layer of regulation on exudates in root tissue against soil-borne microbes in the rhizosphere. The correlation is, however, complex and indirect. Elevated CO_2_ promotes rhizobacterial colonization by *Pseudomonas simiae* WCS417 in Arabidopsis root but not a saprophytic strain *Pseudomonas putida* KT2440 [54]. This happens in nutrient-poor conditions but not nutrient-rich conditions. Therefore, the colonization of soil-borne microbes depends on bacterial species as well as soil quality (nutrients).

### 3.3. Nutrient Status Modulates SA-Dependent Immunity

Nitrogen and phosphorus (P) are the two major mineral nutrients that determine plant growth and productivity. Both nutrients have been implicated in plant defense responses [55,56]. Although it is generally accepted that nutrient status influences disease in plants/crops, much of the findings are contradictory, limiting the application of nutrient/fertilization to facilitate disease control. Two hypotheses shape the studies in determining nutrient–immunity crosstalk. First, pathogen proliferation depends on nutrient availability in planta. Thus, nutrient-limited plants might be better defended. Second, changes in host secondary metabolites in response to external nutrient supply determine plant resistance.

Nitrogen is acquired by plants from the soil as nitrate or ammonium [57], or via symbiotic association with nitrogen-fixing bacteria in legumes [58]. Early study showed that disease susceptibility of tomato (*Solanum esculentum*) depended on external nitrogen supply and was pathogen-specific. Increasing nitrogen (nitrate) supply to tomato promoted susceptibility against the bacterial pathogen *P. syringae* and the powdery mildew *Oidium lycopersicum* in tomato [59]. This observation appears to fit the first hypothesis that pathogens rely on nutrient availability in the plant. Subsequent reports in rice (*Oryza sativa*) [60,61] and wheat (cv. Arche and Récital) [61] using high nitrogen (both nitrate and ammonia) also led to enhanced susceptibility against fungal blast. However, nitrogen supply did not affect plant resistance/susceptibility against the wilt agent *Fusarium oxysporum* f.sp. *lycopersici* [59] nor against *B. cinerea* [62]. Intriguingly, it has been observed that nitrogen content in the apoplast of tomato leaf was further increased after infection [63]. This observation was further supported by a subsequent study to investigate nitrogen management in tomato plants after infection by different pathogens and by chemical elicitation. Glutamine synthetase (GS1) and glutamate dehydrogenase (GDH) regulate nitrogen mobilization in tobacco. Both marker genes were induced by SA, viruses (Cucumber Mosaic Virus, Tobacco Etch Virus, and PVY), bacteria (*P. syringae* pv. *syringae*, *hrp* mutant, and *P. syringae* pv. *tabaci*), and fungal elicitors (cryptogein and Onozuka R10) [64]. Similarly, in common bean (*P. vulgaris*), infection by fungus (*Colletotrichum lindemuthianum*) also activated nitrogen mobilization (*GS1* mRNA) and SA signaling (*PAL3* mRNA) [65]. Interestingly, fungus preferred nitrate as the nitrogen source over ammonium, as tomato plants supplied with ammonium rather than nitrate showed significant reduction in vascular wilt symptoms when infected with the fungal pathogen *Fusarium oxysporum* [66].

The interaction between P and plant immunity, on the other hand, is relatively more evident, as summarized previously [56]. External P supply modulated SA signaling in Arabidopsis [67] and JA-mediated immunity in Arabidopsis [68], tomato (*S. lycopersicum*) [68], tobacco (*N. benthamiana*) [68], and cotton (cv. YZ1) [69]. Internal phosphate starvation responses (PSRs) play critical roles in plant immunity, as mutation of known PSR regulators displayed altered immune phenotype towards different pathogens [70,71,72,73,74,75]. Notably, phosphate starvation response 1 (PHR1) [72], the major transcription factor mediating phosphate starvation response, and phosphate transporters [76,77,78] that modulate P uptake and mobilization, were shown to play a role in the SA-dependent signaling pathway and, hence, plant resistance against various pathogens, highlighting the specific role of P in modulating plant immunity.

The impact of differential potassium (K) fertilization on immunity has also been reported. However, the connection to SA signaling remains fragmented. K deficiency was shown to promote fungal susceptibility in rice (*O. sativa*) [79], while high K supply correlated with enhanced resistance in sweet basil (*Sarocladium oryzae*) [80]. Interestingly, a proteomic study revealed significant downregulation of PR-1 and PR-5 protein in cotton (cv. DP99B) under K deficiency [81], which is in agreement with the immunity results observed in rice and sweet basil.

The major bottleneck for applying nutrient control to achieve protection is compromised growth and, hence, yield. The persistent accumulation of the phytohormone SA under nutrient deficiency might also negatively affect plant growth. However, interestingly, many PSR mutants display constitutive activation of defense signaling, but their fresh weights are not reduced relative to wildtype plants. This could either indicate the existence of alternative transcriptional control to bypass regulation at the hormone level that is sufficient to wall off pathogens, or an important but unidentified nutrient-related molecule that specifically modulates immune signaling without compromising growth.

## 4. Emerging Roles of Development and Primary Metabolism in Defense

Age is a commonly important factor influencing stress response in both animals and plants. Age-related resistance (ARR) was observed in Arabidopsis 20 years ago [82]. Arabidopsis plants at different developmental stages (26–57 days) were challenged with virulent *Pst* DC3000, and there was a clear negative correlation between plant age and bacterial growth. *Sid1*, *sid2* mutant and *nahG* transgenic plants did not show ARR, indicating that SA is required for the ARR pathway [82]. Interestingly, in a subsequent study, ARR was not observed when Arabidopsis plants were challenged with avirulent *P. syringae* pv. *maculicola* (*avrRPM1*) [83]. Moreover, the young leaves exhibited greater SA accumulation and systemic acquired resistance [83]. Young leaves are better energy sources due to having a more-intact photosynthetic apparatus [84]; therefore, they might be better protected. This notion was supported by a more recent study to unveil the crosstalk between ABA and SA. Upon infection with a *Pst* DC3000 *hrcC-* strain, the authors found that young leaves were more resistant [85]. Notably, ABA suppressed SA accumulation and *PR1* gene expression in old leaves, indicating extensive crosstalk between biotic and abiotic stresses, which could be more profound in natural environments.

Development and primary metabolism are the basis for normal cell/plant function. Disruption of key enzymes in developmental and/or metabolic pathways has unexpectedly led to plant immunity findings [86,87,88,89]. The *low expression of osmotically responsive gene 2* (*LOS2*) encodes a cytosolic enolase that catalyzes glycolysis. Knockdown mutants of *LOS2* displayed multiple developmental and growth defects, such as reduced shoot and root growth. It was observed that *los2* mutants exhibited enhanced disease resistance independent of its conventional downstream target, c-Myc binding protein-1 (MBP-1) but related to NLR and SA signaling [89]. Interestingly, metabolomic analysis revealed significant perturbation of glycolysis and sugar accumulation in *los2* mutants. Further, the catalytic activity of LOS2 was required for LOS2 function in both growth and immunity [89]. It will be of great interest to identify which sugar derivative(s) indeed contribute to immune signaling and hence plant resistance. Similarly, disruption of aldehyde dehydrogenase OsALDH2B1 in rice (cv. ZH11) [87], protein phosphatase 2A-B’γ (PP2A-B) [88], and actin-related protein complex 4 (ARPC4) in Arabidopsis [90] also lead to auto-immunity phenotypes, which were different from their conventional roles in growth, fertility, and cytoskeleton formation. Intriguingly, auto-immunity in all these mutants converged at the constitutive activation of SA signaling. Answers to how cells sense perturbations in these developmental and metabolic processes, how they relay these signal to downstream pathways, how they merge all these signals, and why they converge to SA signaling are anticipated.

## 5. Breeding Strategies to Overcome Growth–Defense Trade-Off

Hybrid vigor, or heterosis, is the phenotypic superiority of the F1 progeny over the parents in terms of stature, biomass, and fertility [91]. There are fewer reports investigating the effect of heterosis on defense. Using a combination of Arabidopsis natural accessions, Yang and colleagues explored the role of heterosis in plant defense against *Pst* DC3000 [92]. Interestingly, in a subset of crosses, bacterial growth in the F1 hybrids was significantly reduced versus for their parents. The corresponding induction in *ICS1* and *PR1* transcript levels were also stronger in the F1 hybrids. Consistently, quantification of free SA and SAG has shown increased accumulation of these defense hormone in F1 hybrids, which was abolished by introducing *pad4* mutation, confirming the contribution of SA signaling in the resistant phenotypes [92]. In most cases, the phenotypic superiority of the F1 progeny could be contributed by multiple genes. Thus, sorting out the causal gene might not be straightforward. By analyzing the promoter regions of differentially expressed genes (DEGs) in the F1 progeny, Yang and colleagues further identified *CCA1* as the causal gene leading to enhanced defense in the hybrids [93]. Interestingly, the promoter regions of the DEGs were enriched in the “evening element” (AAAATATCT), which was associated with defense-related and circadian rhythm pathways. The authors thus focused on two known circadian clock mediators, *CCA1* and *LHY* [94,95], for further investigation, and found that disrupting *CCA1* abolished heterosis for both defense and growth superiority in the F1 progeny. The authors thus proposed that CCA1 has different downstream targets during day and night. In the dark, CCA1 promotes SA biosynthesis and enhanced defense, while in the daytime, CCA1 promotes photosynthesis and growth. Thus, heterosis, in principle, can effectively bypasses growth–defense trade-off and achieve both goals simultaneously [93].

## 6. Conclusions and Perspectives

Resolving the interplay between growth and defense under variable environmental conditions will be challenging, but at the same time it is essential to bridge the gap between laboratory research and field application. Based on current information, there is a clear connection between environmental variation and plant immunity (Table 1). Notably, combined stresses might not simply be considered additive of individual stresses. Efforts in studying stress combinations are challenging due to practical difficulties in maintaining similar test conditions, leading to discrepancies between published reports. However, there are indeed common threads, such as phytohormone and related defense signaling, that might serve as the foundation for further research. Interestingly, by focusing on different developmental or metabolic pathways, autoimmune responses do not necessarily follow the entire pathway starting from the hormone, the hormone-dependent transcriptional regulators, and all the way to downstream defense gene activation. Branch points do exist. Transcription factors that determine other metabolic pathways do directly modulate defense gene expression. There are also occasions when growth is not compromised for defense. Altogether, the signaling network that determine growth–defense trade-off is just starting to be revealed. Efforts in this direction are anticipated so that we will be one step closer to engineering the climate-resilient “dream crop”.

## Figures and Tables

**Figure 1 cells-11-02985-f001:**
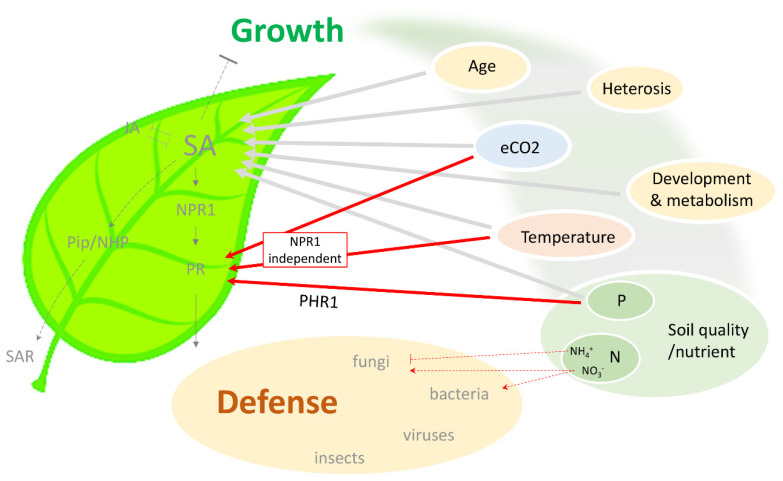
Schematic summary of the connection between different environmental conditions and SA-dependent immunity. Plant growth depends on multiple environmental factors, including temperature, atmospheric CO_2_, and soil quality (nutrient status). These factors have also been found to modulate SA level and SA-dependent defense response against various pathogens. Notably, an NPR1-independent pathway exists in external factor-mediated protection, which potentially bypasses SA and the associated negative effect on plant growth. SA, salicylic acid; JA, jasmonic acid; NPR1, nonexpressor of pathoegenesis-related genes 1; PR, pathogenesis-related; Pip, pipecolic acid; NHP, N-hydroxy-pipecolic acid; SAR, systemic acquired resistance; eCO_2_, elevated CO_2_; P, phosphorus; N, nitrogen; PHR1, phosphate starvation response 1.

**Table 1 cells-11-02985-t001:** Summary of the impact of environmental factors on SA signaling and plant immunity.

Environmental Conditions	Plant Species	Effect on SA or SA Signaling	Immunity Output	Pathogens/Pests	Ref.
* Temperature *					
Low temperature	Arabidopsis	SA and SAG induced	n/a	n/a	[28]
Low temperature	Arabidopsis	SA induced	Promote resistance	*Pst* DC3000	[29]
Low temperature	Arabidopsis	SA induced	Promote resistance	*Pst* DC3000, *Cor-*, *hrcU-*	[30]
High temperature	Arabidopsis	n/a	Promote susceptibility	TMV	[3]
High temperature	tomato (*S. lycopersicum*)	n/a	Promote susceptibility	TMV	[31]
High temperature	tomato (*S. lycopersicum*)	n/a	Promote susceptibility	TYLCV	[32]
High temperature	potato (*S. tuberosum*)	SA signaling more active	Promote susceptibility	PVY	[33]
		in resistant cultivar			
* Atmospheric CO_2_ *					
eCO_2_	Arabidopsis	SA induced	Promote resistance	*P. cucumerina* and *H. arabidopsidis*	[39]
eCO_2_	Arabidopsis	SA induced	Promote resistance	*Pst* DC3000 and *B. cinerea*	[40]
eCO_2_	beans (*P. vulgaris*)	SA induced	n/a	n/a	[40]
eCO_2_	wheat (*T. aestivum*)	SA induced	n/a	n/a	[40]
eCO_2_	tomato (*S. lycopersicum*)	SA induced	Promote resistance	TMV, *P. syringae*, *B. cinerea*	[41]
eCO_2_	maize (*Z. mays*)	no change	Promote susceptibility	*F. verticillioides*	[42]
eCO_2_	wheat (cv. *Remus*)	n/a	Promote susceptibility	*Z. tritici*	[43]
saCO_2_	Arabidopsis	SA induced	Promote susceptibility	*P. cucumerina*	[39]
saCO_2_	Arabidopsis	SA induced	Promote resistance	*H. arabidopsidis*	[39]
* Nutrient *					
High nitrate supply	tomato (*S. esculentum*)	n/a	Promote susceptibility	*Pst* DC3000	[59]
High nitrate and ammonia supply	rice (*O. sativa*)	n/a	Promote susceptibility	*M.oryzae*	[60]
High nitrate and ammonia supply	rice (*O. sativa*)	n/a	Promote susceptibility	*M.oryzae*	[61]
High nitrate and ammonia supply	wheat (cv. Arche and Récital)	n/a	Promote susceptibility	*M.oryzae*	[61]
Ammonium supply	tomato (*S. lycopersicum*)	n/a	Promote resistance	*F. oxysporum*	[66]
Low P supply	Arabidopsis	SA induced	n/a	n/a	[67]
Low P supply	tomato (*S. lycopersicum*)	n/a	Promote resistance	*S. littoralis* and *P. brassicae*	[68]
Low P supply	cotton (cv. YZ1)	n/a	Promote resistance	*V. dahliae*	[69]
Low P supply, *phr1*	Arabidopsis	SA signaling more active	Mutant more resistant	*Pst* DC3000	[72]
*pht4;6*	Arabidopsis	SA and SAG induced	Mutant more resistant	*Pst* DC3000	[76]
*pht4;1*	Arabidopsis	suppresses SA in *acd6-1*	Mutant more resistant	*P. syringae* pv. *maculicola*	[77]
Low K supply	rice (*O. sativa*)	n/a	Promote susceptibility	*S. oryzae*	[79]
Low K supply	cotton (cv. DP99B)	PR-1, PR-5 repressed	n/a	n/a	[81]
High K supply	sweet basil (*O. basilicum*)	n/a	Promote resistance	*B. cinerea*	[80]
* Development and primary metabolism *					
Age related resistance (ARR)	Arabidopsis	n/a	*sid1*, *sid2*, *nahG* abolish AAR	*P. syringae* pv. *maculicola*	[83]
*los2* (glycolysis)	Arabidopsis	SA induced	Mutant more resistant	*Pst* DC3000	[89]
*osaldh2b1*	rice	SA induced	Mutant more resistant	*X. oryzae* and *M. oryzae*	[87]
*pp2a-b*	Arabidopsis	SA induced	Mutant more resistant	*B. cinerea*	[88]
*arpc4*	Arabidopsis	SA signaling more active	Mutant more resistant	*S. sclerotiorum*	[89]
* Breeding strategy *					
Hybrids/ heterosis	Arabidopsis	SA induced	Promote resistance	*Pst* DC3000	[92]
Hybrids/ heterosis	Arabidopsis	SA induced	Promote resistance	*Pst* DC3000	[93]

n/a, not available or not applicable; eCO_2_, elevated CO_2_; saCO_2_, sub-ambient CO_2_; TMV, tobacco mosaic virus; TYLCV, tomato yellow leaf curl virus; PVY, potato virus Y; AAR, age-related resistance.

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
