# Peer review of "Progress in Salicylic Acid-Dependent Signaling for Growth–Defense Trade-Off"

_cells, 2022, doi:10.3390/cells11192985_

Round 1

Reviewer 1 Report

This paper is a condensed and well-written review on the role of SA and its signaling  under multiple stress factors, especially on the growth-defense trade-off.  The paper covers an interesting subject but has some questionable points for the reader, listed below. Therefore this paper suggested for publication after the necessary  re-considerations and/or explanations.

Abstract

The first sentence "One grand challenge for studying plant biotic and abiotic stress responses is to optimize plant growth and plasticity under variable environmental constraints."  There is not only a problem for the scientist during experiment, but in fact a problem for the practical agricultural production to overcome yield losses!

L 14 upon the perception of pathogens and elicitors. Pathogens are percepted mostly by elicitors of pathogen origin and indirectly of plant origin induced by pathogenic enzyme activities.

Introduction

L 22-23 Only two environmental abiotic stressors are listed (temperature and carbon-dioxide enrichment). The latter untill now usually is not listed as a stressor. Therefore it seems a little bit  striking that both water shortage and flooding (two important factors nowdays with climate change) is absent. See also Table 1. and Fig. 1  from this point of view.

Table 1 Check the correct spelling of P. syringae pv. Maculicola.

Table 1, There is no nitrate supply data, only ammonia, but Fig 1 lists nitrate as N-source but the reason is not completely clear. See also p. 7 for nutrients.

L44-55 HORMONES are listed, but ABA and especially N-hydroxy-pipecolic (NHP) acid are absent. For further consideration,  not only SA-glycosylation but NHP-glycosylation play an important role between  SAR-response and plant growht reduction.  See Cai et al., 2021, Molecular Plant 14, 440 and other papers.

L 61, Fig. 1. SAR is indicated as an SA-signalled/mediated (? it is not clear) process, but in fact the signal is not SA, but rather Pip or NHP (see Attaran et al., 2009 and other papers of Zeier group)  and both de novo NHP and NHP-mediated SA synthesis is required in upper leaves for resistance induction.

Temperature

p. 5,  L 134"The bacterial resistance phenotype was eventually confirm by challenging Arabidopsis with the bacterial pathogen Pseudomonas syringae pv. tomato strain DC3000 (Pst DC3000) [29,30]." The effect on bacterial titer was attributed only to SA-accumulation and the direct effect of temperature was excluded.

L 144 cold stressed-induced protection. cold stress-induced?

Please, if possible make distinction between direct effect of cold treatment and vernalization (when the effect of the acclimatization process is stuidied later on under normal conditions).

Nutrients

p.7 L255  The reasons for the nutrient preference in the fun
gus, and the plants’ decision to promote nitrogen remobilization under pathogen attack remain obscure.
Ammonia seems more toxic than nitrate and amino acid metabolism is fully different.

Reviewer 2 Report

This manuscript is an interesting review, that has a good written and brings good information about this subject.  I think this manuscript is almost complete and good for publish, however I have some highlights for typing mistakes in the text (see attached file) and I think in the topic about Nutrient status modulates SA-dependent immunity, the authors can could writing about the low and high nitrogen effect in the Blast disease in rice, and the importance of the Calcium – Potassium balance in the plant pathogen interaction, manly in the plant resistance to diseases caused by Cercospora fungi. Potassium is important in many PR Proteins that’s correlation with salicylic acid signaling.

After add topics I think this review can have condition to be accepted to publish.

Round 2

Reviewer 1 Report

Most issues were properly corrected.